# Analysis of MATSim Modeling of Road Infrastructure in Cyclists' Choices in the Case of a Hilly Relief

Younes Delhoum [ID], Rachid Belaroussi *[ID], Francis Dupin [ID] and Mahdi Zargayouna [ID]

COSYS-GRETTIA, Université Gustave Eiffel, F-77447 Marne-la-Vallée, France
* Correspondence: rachid.belaroussi@univ-eiffel.fr

**Abstract:** For too long, many refined transportation models have focused solely on private and public transportation, assuming that bicycles only require simple models, such as bird flight distance or trips on horizontal tracks at a constant speed. This paper aims to study the impact of the road characteristics, such as road gradient, type of road and pavement surface of the road, on cyclists' behavior using dedicated modules of MATSim. For that, we compare two approaches: a standard approach which does not consider the road characteristics, and a second approach that uses MATSim bicycle extension of Ziemke et al. The two approaches are analyzed over a sub-regional area around a district, focusing on a suburban city with an undulating relief made of average-to-steep hills. The focus is on the bicycle transportation model because the catchment area has a particularly challenging altitude profile and a large variety of roads, whether in type—from residential to national highway— or in pavement surface due to the number of green areas, such as parks and forests. This area is defined as a rather large 7 × 12 km, including five suburban cities in the South of Paris, France. A synthetic population of 126,000 agents was generated at a regional scale, with chains of activity made of work, education, shopping, leisure, restaurant and kindergarten, with activity-time choice, location choice and modal choice. We wanted to know how accurately a standard model of bicycle travels can be made with a 2D flat Earth assumption by comparing it to an algorithm extension that explicitly considers road characteristics in cyclists' route choices. Our finding is that the MATSim bicycle extension model impacts mainly the long trips. Otherwise, the differences are minimal between the two models in terms of travel time and travel distance.

**Keywords:** cycling infrastructure; MATSim; bicycle simulation model; cycling route choice; activity-based model; synthetic population

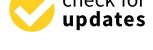



## 1. Introduction

Cycling is gaining ground as an inexpensive, healthy, energy-efficient and fast transport mode for trips shorter than 5 km; it is increasingly used notably in urban regions and can be profitably associated with public transport [1,2]. Integrating this trip mode into transportation models is key to helping the planners and decision-makers of transport systems.

The activity-based modeling (ABM) framework was originally developed for more realistic travel demand models. These models can analyze a wider range of transportation policies and evaluate travel demand and transportation supply management strategies.

In this study, the activity-based transport simulation framework MATSim is used. It requires a transport travel demand represented by a synthetic population which includes daily plans (each plan is a sequence of activities and legs), a transportation network, and a public transport scheduling [3].

The default modeling of bike trips in MATSim is *teleportation*: a synthetic traveler is moved directly from a previous activity location to the next activity location, with a constant travel speed and without any network interaction. This approach is unrealistic and does not reflect the behavior of cyclists who move using the infrastructure [4].

Dobler et al. [5] implemented an approach to obtain more realistic travel speeds for cyclists compared to teleportation by considering traveler attributes, such as age, gender and link slope. However, this approach does not explicitly simulate cycling agents on the network (no congestion). It also does not consider the cycling-relevant characteristics of the infrastructure, such as the gradient and nature of the road.

Ziemke et al. [4] proposed a bicycle model by considering the characteristics of the infrastructure and the interaction of cyclists with motorized traffic. This extension contains two main components: a speed calculator, which calculates the travel speed of the cyclist by link and depends on the quality of the road and its slope; and a cycling itinerary calculator, which selects the trip route based on the travel distance, travel time, type and gradient of the route. Table 1 gives an overview of the main characteristics of the bicycle models in use in MATSim.

**Table 1.** MATSim models of bicycle trips: with or without traffic congestion and characteristics of the road.

| MATSim Bicycle Models | Cong. | Characteristics |
|:---:|:---:|:---:|
| Teleportation<br>Axhausen et al., 2016 [3] | No | - |
| Multimodal extension<br>Dobler et al., 2016 [5] | No | Road slope |
| **MATSim bicycle standard model**<br>Axhausen et al., 2016 [3] | Yes | - |
| **MATSim bicycle extension model**<br>Ziemke et al., 2019 [4] | Yes | Type of road<br>Pavement surface<br>Road slope |

In this paper, we propose to integrate cyclists in MATSim in a realistic way—with trips on the road network instead of teleportation, with or without the additional Ziemke module. We provide an analysis of the impact of road characteristics on the cyclists' behavior and the accuracy of simulations integrating this information.

## 2. Literature on Bicycle Travels

There is a body of research about bicycle travel modes to understand the dynamics of bicycle commuting. Most studies focus on the road environment, cyclist infrastructure, and cyclist and driver behavior. Their scope extends to the human with bicycling attitudes, the vehicle with the onboard sensing of bicycle dynamics, and technology regarding electric versus mechanical or built environment. An individual-centered study can investigate cyclists' stated or revealed preferences, cyclists' practices, users' experiences and sometimes across cycling cultures. Sensors-based studies bring a Lagrangian point of view on bicycle traffic flow. It is also useful for bike sharing systems [6] and rebalancing operations. Counting station-based measurements bring an aggregated Eulerian point of view on bicycle traffic flow and the share of bicycle commuters in transportation. They are analyzed for bike traffic estimation and forecast, market trends, sustainable transport management recommendations for cities and transport service operators, or bicycle infrastructure assessment.

The COVID-19 pandemic has enormously impacted travel behavior in most of the world. Buehler et al. [7] examine the available evidence about the impact of the pandemic on cycling in various cities and countries and overall new trends in cycling. The usage of traditional bicycles, as well as e-bikes and other personal mobility devices, has grown tremendously, especially with the recent market of bike-sharing systems.

Some studies focus more on safety aspects of cycling: Salman et al. [8] present the findings from a literature review on bicycle crash contributory factors, whereas Leng

et al. [9] review recent developments and innovation of the bicycle helmet design. One major challenge of bicycle safety is the lack of complete exposure data.

The bicycle can be a convenient transport mode. However, its benefits decline with increasing distance: Banerjee et al. [10] give a literature review on what stimulates bicycle commuting beyond 5 km by analyzing socio-psychological and physical factors. Perceived trip benefits, cycling habits, bicycle-friendly infrastructure, and e-bike usage are identified as key factors.

To help with the planning and operation of bicycle transportation systems, Bai et al. [11] explore factors that affect bicycle usage levels of a dock-based bike sharing system. Levels of bicycle usage are defined based on cycling frequency, duration, and time of day. Influencing factors vary with land-use patterns of residential and workplace locations.

Given the structural complexity of transportation networks, cycling network development often follows a slow and piecewise process. Szell et al. [12] systematically explore the topological limitations of urban bicycle network development, with different variations of growing a synthetic bicycle network between an arbitrary set of points. Poliziani et al. [13] use GPS traces in an algorithm that estimates travel times from map-matched GPS traces and associates them with infrastructure attributes. Other sources of data are the cyclists' stated preference [14,15].

In cyclist route choice modeling, the traditional approach uses external factors and generates choice sets of paths with a logit model to create a discrete route choice model. A recent work by Magnana et al. [16] uses deep and machine learning algorithms on GPS tracks to learn representations from the data, which replace explicit factors.

Other topics of interest related to electric bikes are an individual vehicle or a fleet in bike sharing systems. Matyja et al. [17] use sensors, such as a GPS module and the barometric altimeter, to estimate the trajectory, the distance traveled, and the height of the route above sea level and its slope. The study aims to determine whether the data can be useful for scientifically evaluating the cyclist–electric bicycle system. The study is completed in [18] with speed sensors and data from the cadence sensor, power measurement, pedaling technique and heart rate to assess the energy efficiency of electric bicycles and intelligent power management systems. The two studies highlight the advantages and drawbacks of onboard sensors as tools for measuring and acquiring data.

Turon et al. [19] have developed a model of factors that influence the operation of shared electric mobility by incorporating aspects related to the COVID-19 pandemic and its impact on this industry. They identify the main factors influencing the electric shared mobility industry during the COVID-19 and post-lockdown periods.

More generally, smartphone tracking applications and bike-sharing systems data provide opportunities to estimate territory-wide bicycle activities. However, due to the urban canyon effect or low data acquisition frequency, the obtained ridership suffers inherently from under-reporting and significant localization errors. Furthermore, big data are not always available, or they are in an aggregated format at a regional scale, making inferences on bike traffic at the mesoscopic level (i.e., lane level) very unreliable.

Therefore, conventional travel-diary censuses are still required unless researchers own a fleet of bikes with onboard sensors. Consequently, simulations with dynamic traffic assignment models integrated with activity-based models provide valuable tools for estimating route choices and bike traffic flow at a regional scale.

## 3. Context: A Real Estate Development in a Hilly Relief

LaVallée is a large construction project recently started in Paris's suburbs. This future district's spatial extent is about 500 m × 400 m, part of a larger city called Châtenay-Malabry. The whole district is to be delivered in 2024. As of July 2022, the first phase is under completion, and the first inhabitants have just started to move in. Figure 1 illustrates the localization of the district and its suburban environment. It will house 6200 residents (20% of the city population), a large set of services ranging from a shopping street, a mall, schools

and office spaces for 2000 employees; the district is designed to be open to the rest of the city.

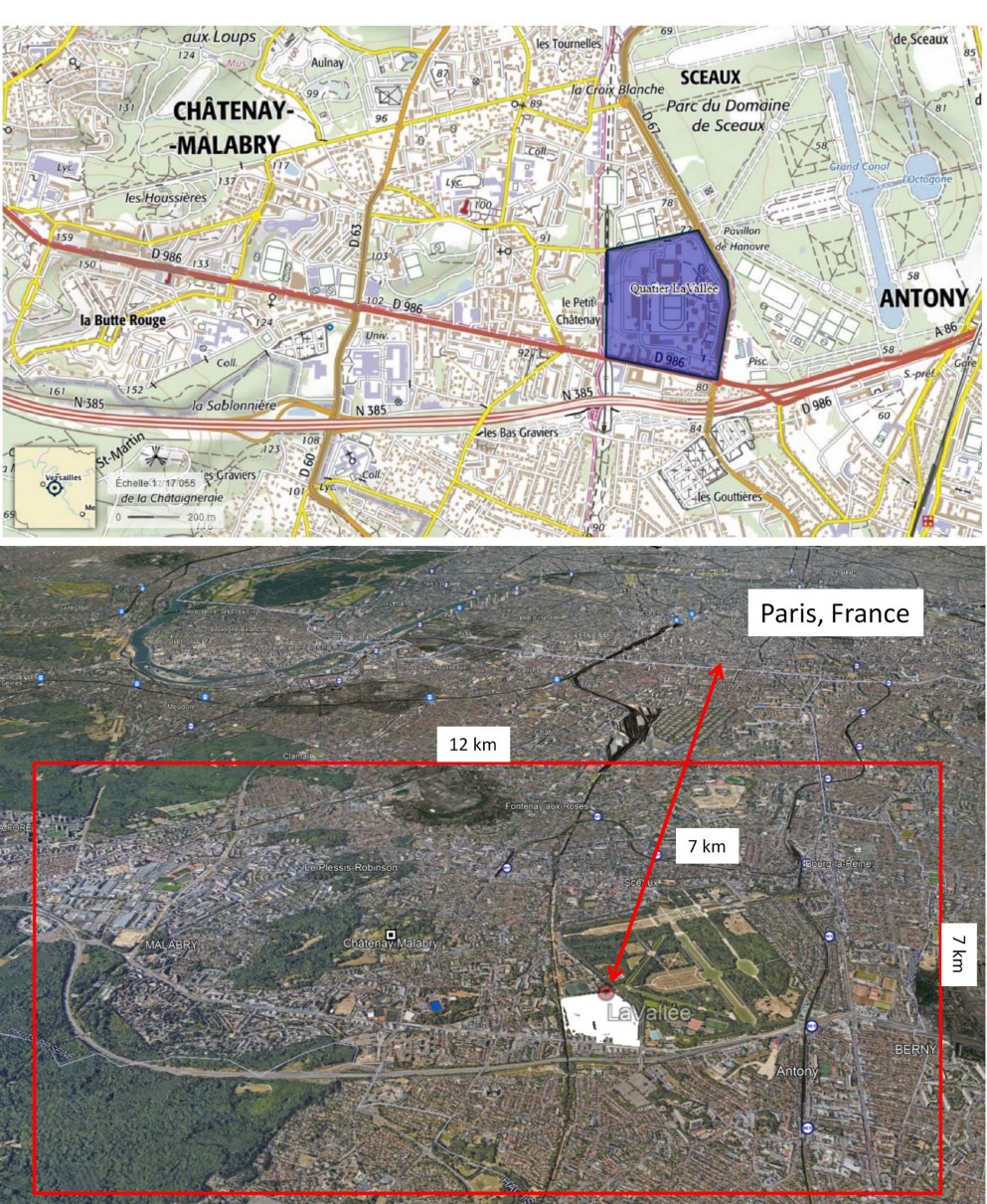

**Figure 1.** LaVallée district and its catchment area, a 12 × 7 km suburban area of Paris. Source: OpenStreetMap and Google Earth.

Eiffage company, the real estate developer, and our university are co-funding a research project on the sustainable development of cities called E3S (EcoQuartier Smart, Sustainable and Secure). University Gustave Eiffel contributes to E3S by developing a living lab of micromobility and numerical methods to predict potential visits and places of interest based on mobility scenarios in future real estate. Based on the collection of traces left by digital ecosystems, our overall objective is to develop a method that makes it possible to model the potential visits of the various equipment and public spaces of the project by mobilizing data from the census at the departmental level and the layout of shops and future activities inside LaVallée.

This predictive model needs to consider the flow of external visitors, estimated realistically based on the pre-project movements in the areas of influence of LaVallée, hence a travel model of the current situation. We built a dynamic travel model around the city of

Châtenay-Malabry in a reference baseline situation without the district. Activity plans of the population are described from available census data.

We defined the catchment area of LaVallée district as the area that potential future visitors can reach. Potential visits correspond to a need for a specific purpose (shopping, leisure, and school) that is currently fulfilled outside of the district but would be more convenient if realized in LaVallée, when the real estate is completed (two years from now).

Figure 1 indicates the location of the district of interest and the catchment area where LaVallée would attract a population that will use its services. This area is defined as being rather large, including not only the whole city, but also the surrounding cities because it will be accessible by private car or public transportation in less than 15 min.

We presented in [20,21] a method for synthetic population generation focusing on activity–time choice, location choice and modal choice at a regional scale. We showed that our method could construct plannings for 126k agents over five municipalities, with chains of activity made of work, education, shopping, leisure, restaurant and kindergarten, which adequately fit real-world time distributions.

In this paper, we focus on the cyclist transportation model because the catchment area has a particularly challenging altitude profile and a large variety of roads, whether in type—from residential to national highway—or in pavement surface due to the number of green areas, as shown in Figures 2 and 3.

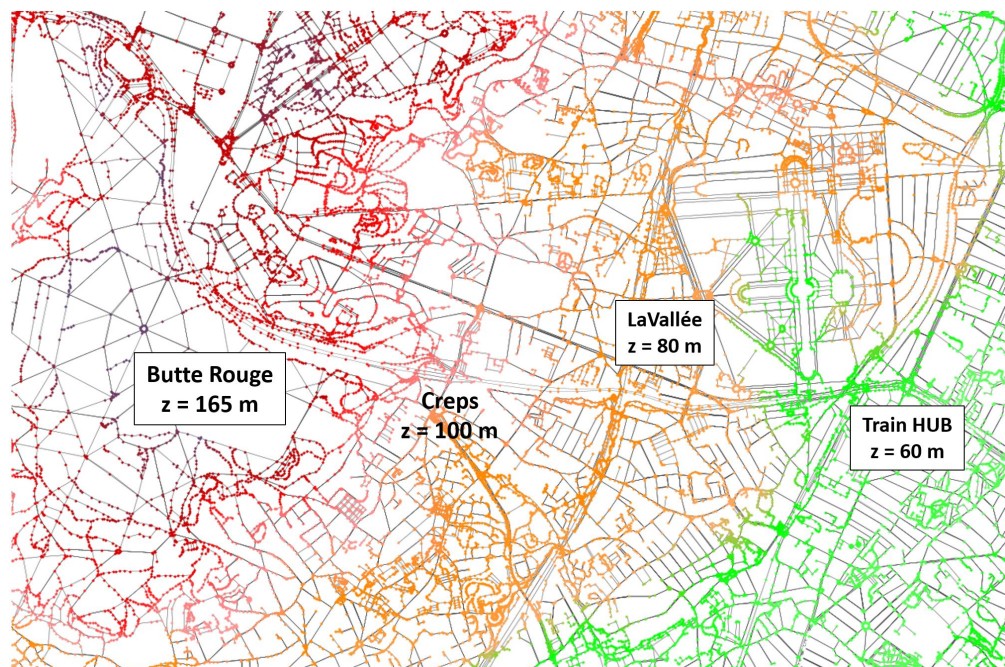

**Figure 2.** Altitudes from sea level. The city spreads over an area with three main plateau. The train station is the main hub of the area to get to Paris, while LaVallée and Butte Rouge are two important bases of population.

The transportation simulation we used is an activity-based model with dynamic traffic assignment: during the day, an agent participates in different activities with predetermined location and purpose, and the choice of departure time and mode of transport is estimated relative to the traffic condition, personal routing behavior and the individual possibility to adapt their hours.

The synthetic population represents a realistic population in an artificial environment. An activity-based model uses this population to generate an activity plan for the different individuals describing their daily journey. A plan comprises a set of activities in which the individual will participate, and a set of legs representing its trips between two activities. Each activity has a start time, end time, and duration that need to be set, as well as its location for each individual.

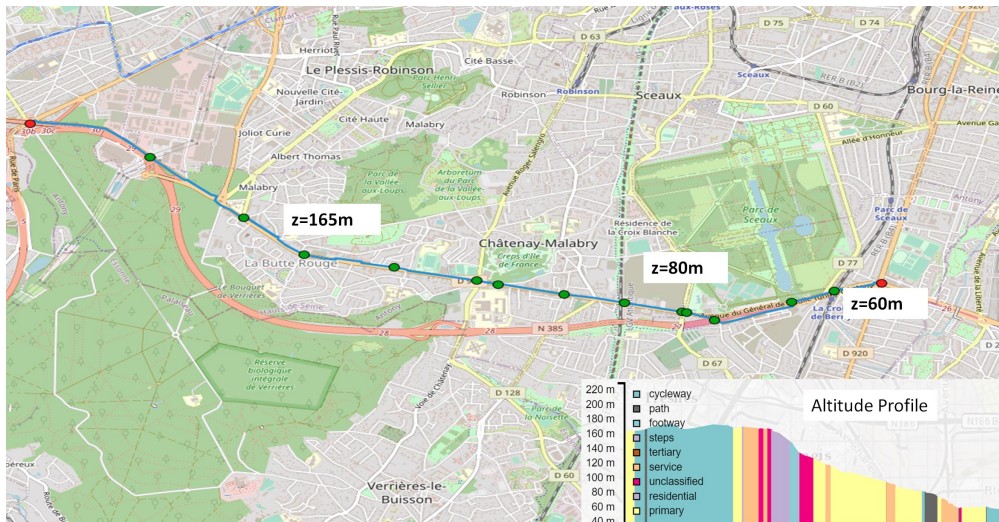

**Figure 3.** Altitude profile over the blue path across the city, with the various types of road. Source: https://www.maps.ie/map-my-route/(accessed on 18 August 2022). Cartography: OpenStreetMap.

We use the MATSim (multi-agent transport simulation) framework for the demand modeling and agent-based mobility simulation: it is open source and widely used to implement large-scale activity-based modeling. Trip purposes included in this study are work, education, shopping, leisure, restaurant and kindergarten. Transportation modes are private vehicle (car), public transportation, walk, and bicycle. Public transportation is made of three modes: buses which are assigned on the car network, railway, and tramways.

We analyze mobility patterns over a sub-regional area large enough to consider all potential visitors. That region includes all inhabitants able to access the district in 15 min, whatever the mode of transportation, considering restrictions related to the built environment. Elements structuring the population mobility are integrated, such as the presence of a highway in the south, which acts as a barrier to the possibility of trips toward or out of the city. The planned tramway line is also a structuring element in the future mobility pattern; the catchment area is spread around it. A total of 126,000 people were selected to constitute the agents of a synthetic population representing the resident of this area.

This study analyzes the agents using bicycles for their activity in this challenging area. It is a study on a recent MATSim module developed to better model cyclists' transportation: it considers road grades, road type and pavement surface. The model is assessed and sometimes criticized in light of the modification in cyclists route choices it outputs, compared to a standard 2D model that does not include the infrastructure characteristics.

## 4. Methodology

### 4.1. Replanning Routes of Daily Plans of a Population

Consequently, based on the first work estimating the travel patterns of agents realistically over a large suburban region [21] based on census, built environment and facilities data, a synthetic population is generated. The daily plans of all population agents are devised with the following activities: work, education, leisure, shopping, restaurant, and kindergarten. Transportation modes included car, public transportation, bicycle, and walk. This study focuses on cyclists, so we extract the planning of agents using a bicycle at least once in a leg to and from an activity: Therefore this study does not cover the effects of mixed traffic bicycles/cars/pedestrians.

Two simulations are run separately, one with the standard model and one with the extension. Each simulation is many iterations of planning schedule executions during a day. Schedule executions consist of three steps: mobility simulation, scoring and replanning. Mobility simulation is a mesoscopic model of traffic. Scoring gives a score to the plans of each agent according to the utility of the routes it took. Replanning occurs at the end of the day, where alternate routes or planning can be chosen to increase the plan's score.

The standard model replanning is mainly based on a travel time proportional to the travel distance of the route taken to accomplish daily planning. With the extension model, the cyclists' speed and riding comfort vary depending on the nature of the links used, affecting the score of their plans. This enables a rerouting decision toward more cycle-friendly ways and alternate routes *a priori* longer than the one used in the standard model.

After completion of the two simulations, they are compared in terms of changes in routes taken—according to the type of route, surface and slope—trip duration and cumulative positive elevation. The question is how large the error is when considering a flat Earth-type simulation.

*4.2. Scoring Functions in a Daily Travel*

A MATSim network consists of a set of nodes and links. A node stores the geographical coordinates, while a link represents a section of the road between two nodes. We collected a 2D network from OpenStreetMap [22], which contains information about roads, including length, speed limit and flow capacity. The elevation of nodes and the slope of links are extracted from an elevation database (European Digital Elevation Model), which is public and has a resolution of 25 m [23].

The daily plans of travelers are generated using a synthetic population process; in this study, the plans are generated using the model that we described in [20,21] and are used to create the simulation scenario.

The MATSim [3] process consists of three modules: mobsim, scoring and replanning. The mobility simulation module executes the selected plans of the whole population in parallel. The scoring process is applied to evaluate the quality of the executed plans using a utility function; it assesses each activity and leg of the plan. In the replanning stage, the traveler selects (or modifies) a plan to execute in the next iteration of the process. The process is repeated until convergence (no traveler can improve their score further).

In the scoring process, the agent calculates the score of the executed plan based on the simulation and the concept of utility. The score is composed of two sums: $S_{act,q}$ utility of performing an activity $q$, and $S_{trav,q}$ dis-utility of making the trip (leg) to the activity $q$. $N$ represents the number of activities in the plan, and the score function is defined in Equation (1).

$$S_{plan} = \sum_{q=0}^{N-1} S_{act,q} + \sum_{q=0}^{N-1} S_{trav,q} \tag{1}$$

A standard approach to model the bicycle traffic in MATSim is by defining a bicycle vehicle with similar behavior to a car with a travel dis-utility function defined by Equation (2).

$$S_{trav,q}^{std} = C_{bicycle} + \beta_{trav,bicycle} \cdot t_{trav,q} + \beta_{d,bicycle} \cdot d_{trav,q} \tag{2}$$

where $C_{bicycle}$ is a bicycle-specific constant, $\beta_{trav,bicycle}$ ($\beta_{d,bicycle}$) is the direct marginal utility of trip duration (distance) when traveling by *bicycle*, and $t_{trav,q}$ ($d_{trav,q}$) is the travel time (distance) on leg $q$. The marginal monetary costs of traveling by bicycle are supposed to be negligible. Agents use this score in the replanning step of the loop: which plan, route or time choice to execute in the mobility simulation of the next iteration.

In [4] the travel dis-utility function is extended as follows:

$$S_{trav,q}^{ext} = S_{trav,q}^{std} + \sum_{a \in q} (\beta_{inf(a)} + \beta_{comf(a)} + \beta_{grad(a)}) \cdot l_a \tag{3}$$

where $a$ is a traversed link, $\beta_{inf(a)}$ is the marginal utility of distance on the infrastructure of $a$, $\beta_{comf(a)}$ is the marginal utility of distance on the level of comfort of $a$, $\beta_{grad(a)}$ is the marginal utility of distance on the gradient of $a$, and $l_a$ is the length of $a$. The marginal monetary costs of traveling are negligible for cycling. The travel time $t_{trav,q}$ takes into account the speed of a cyclist on a given link.

This paper aims to study the impact of road characteristics, mainly the slope, type and surface of roads as described in Table 2, on cyclists' behavior. For that, we compare two approaches: a standard approach that does not consider the road characteristics and uses Equation (2) as a scoring function, and the second approach which uses the MATSim bicycle extension with Equation (3).

**Table 2.** Road types and surfaces.

| Infrastructure | OSM-Tag Value |
| --- | --- |
| Type of road ("highways") | Cycleway, primary and secondary, residential |
| Pavement surface | Asphalt, gravel, sand |

*4.3. Some Insights of the Bicycle Extension Model*

As mentioned earlier, the travel time $t_{trav,q}$ takes into account the speed of a cyclist on a given link. The MATSim bicycle standard model does not consider the impact of the road characteristics on cyclists' behavior: speed only depends on traffic density. Since we are not considering motorized traffic in our simulation, the cyclists' velocity is free-flow speed. When computing travel times, the maximal speed by which a bicycle can traverse a link is constant in the standard model, set at 20 km/h.

The extension model considers the effect of road characteristics as follows:

- The type of infrastructure affects the velocity of the cyclist;
- Pavement surface can mitigate both the riding comfort and the velocity of the cyclist;
- Road gradients impact the velocity of cyclist: speeds are decreased linearly with an increasing slope so that a 10% slope decreases the speed by half [24]; Downhill speed increases are not considered in the extension model because of their lower impact on cyclists' choices.

Overall, road characteristics impact the travel speed of a cyclist based on the proposed velocity factors in Table 3. The dis-utility factor $F_{inf}$ for the infrastructure reflects how the type of road impacts the cyclist in their trip on a link *a*. He/she can travel at full speed in a cycleway, whereas the speed can be consistently reduced on the primary road for motorized vehicles. The dis-utility factor $F_{comf}$ relates to the comfort of riding experience on asphalt superior to a graveled or sandy link, reducing travel speed. These speed reduction values are taken from [4].

**Table 3.** Velocity and dis-utility factors.

| Type | Speed | $F_{inf}$ |
| --- | --- | --- |
| Cycleway | 100% | 0% |
| Residential | 85% | 15% |
| Secondary | 30% | 70% |
| Primary | 10% | 90% |
| **Surface** | **Speed** | $F_{comf}$ |
| Asphalt | 100% | 0% |
| Gravel | 60% | 40% |
| Sand | 30% | 70% |

The marginal dis-utility values are chosen as follows: $\beta_{trav,bicycle} = -6.0$ utils/h, $\beta_{d,bicycle} = -0.0004$ utils/m, as mentioned in [4].

Parameter $\beta_{inf(a)}$ reflects on the existence of a seamless cycling infrastructure in a given link a; it dependent on the highway type and is computed as $\beta_{inf(a)} = -0.0002 \cdot F_{inf}$ utils/m.

The marginal utility of comfort $\beta_{comf(a)}$ depends on the riding smoothness with a bicycle: the smoother the pavement, the higher the utility. It is computed as $\beta_{comf(a)} = -0.0002 \cdot F_{comf}$ utils/m.

Based on the marginal rates of substitution [25], an average cyclist will prefer a detour of 500 m rather than climbing a hill 10 m in height, the marginal utility of distance on the gradient of link a is chosen as $\beta_{grad(a)} = -0.0004$ utils/m.

## 5. Experiments

### 5.1. Case Study

This work is the continuation of the works started in [20,21], where we modeled the travel behaviors of the population of a catchment area (126,151 agents). The input data used to generate the synthetic population are as follows:

- BPE (Base Permanente des Équipements): Contains information about the different facilities' locations and types. It provides the equipment and services rendered by the territory to the population.
- INDCVI (Individu Localisé au Canton ou VIlle): Contains individual information, such as socio-demographic and household characteristics.
- MOB-PRO (Mobilité Professionnelle): Bi-localized home-work trip data describing the individual's characteristics, household and main residence. The data are sorted by residence-city and work-city. It is related to working individuals aged 15 years or more.
- ENTD (Enquête Nationale Transports et Déplacements): Census on trips of households residing in metropolitan France and their use of both public and individual modes of transport. This survey describes all trips, regardless of the reason (origin and destination of the activity), distance, duration, mode of transport used, time of year, or time of day. It is used to extract the following data distributions: (1) activity end-time, (2) activity duration, and (3) the travel speed of each mode of transport.

The focus of this case study is on bicycle trips. Cyclists represent 8% of the population. They represent 3% of the total trips. The distribution of trip purpose is as follows: 34% for work, 32% for leisure, 25% for shopping, 5% for restaurant and 4% for education, as shown in Figure 4. Most bicycle trips are within a radius of 5 km and a travel time under 20 min.

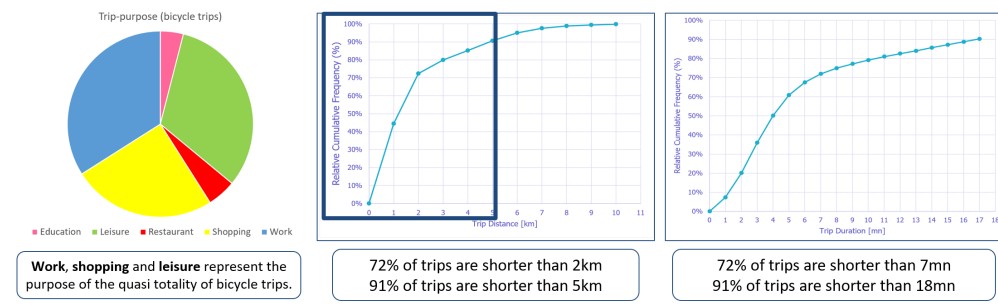

**Figure 4.** Statistics about bicycle usage in the study area: activities, travel distance and travel time.

The study area is spread over five suburban cities and is shown in Figure 5. The area was chosen to correspond to the accessibility of 15 min. The area's relief is highly undulating, with three plateaus starting at an altitude of 50 m from sea level to 200 m. Three-quarters of the network is on the level of terrain (slope lower than 1%), a tenth of routes have a negligible slope (slope between 1% and 3%), and 7% of the routes have a slope between 3% and 6%. The remaining routes are steep and represent 8% of the bicycle network. More details are shown in Figure 5.

The study area network is a multi-modal network. It consists of 83,176 nodes and 321,145 links, including 88,049 bicycle links. A total of 75% of the bicycle network provides favorable and fast routes for cyclists. These routes are mainly residential, cycleways and paths. A large portion of 25% of network routes disadvantage the cyclists. They are mainly primary and motorway routes, as can be shown in Figure 6.

In terms of pavement surface, the bicycle-network routes are distributed between asphalted (89%) and non-asphalted (11%); the non-asphalted routes consist of gravel (8%) and sand (3%), and they are mainly located in the green spaces. The proportion of cobblestone routes is negligible (around 0.1%). A more detailed view of the pavement surface of the network can be found in Figure 6.

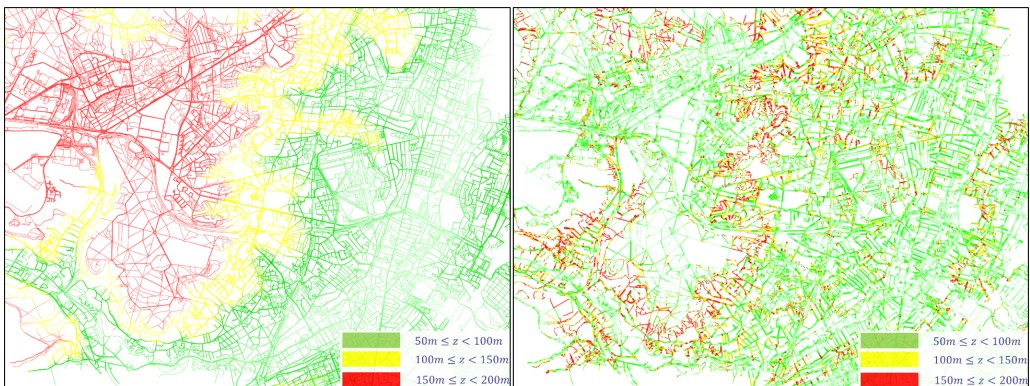

**Figure 5.** (**Left**): Elevation map: from 50 m to 200 m from sea level. (**Right**): bicycle roads slope.

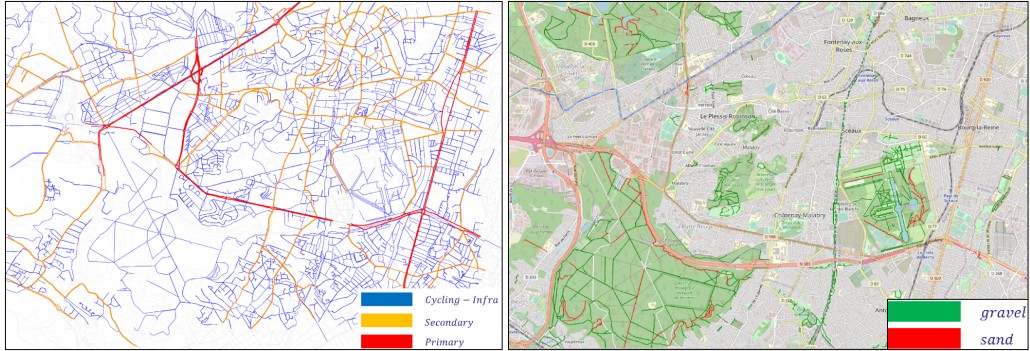

**Figure 6.** Type and pavement surface of bicycle infrastructure in the study area. The simulation uses a synthetic population of 9566 cyclists (8% of the whole population of 126,151 agents). (**Left**): primary highways (red) and secondary highways (orange) are prohibited (or to avoid) for cyclists, and the remaining routes are mainly residential (blue) cyclist friendly. (**Right**): gravel and sand routes are concentrated in the green spaces, and the remaining routes are considered asphalt.

### 5.2. Simulation Results

Activity plans for the whole population of the catchment area (126,151 agents) are generated. The focus of the case study is on bicycle trips. Once the plans are generated, the plans with a bicycle trip are selected and then simulated. The total number of agents (9566) represents around 8% of the whole population of the catchment area but only 3% of the total trips.

Two simulations are run: a simulation with a standard approach using Equation (2) as a scoring function, and a simulation with the second approach using the MATSim bicycle extension with Equation (3).

The concurrent simulations show that the use of the road surface is quasi-identical to both standard and extended score functions as shown in Figure 7: 98% of cyclists' trips are on asphalted routes. The traffic flow is well distributed between the large highways and the cycling-friendly infrastructure with the standard score, while it is reduced by 7% with

the extended score. In the standard scenario, 15% of bicycle traffic flow passes by primary highways, and 33% passes by secondary highways, while 52% passes by a cycling-friendly infrastructure; in the extension scenario, it is noticed that the traffic is reduced by 4% on primary highways and by 3% in the secondary highways, while it is increased by 7% in the cycling-friendly roads, compared to the first scenario.

The third factor is the gradient of the route. The simulation shows a negligible gap between the two scenarios (around 1%). This small difference is the consequence of avoiding the steeper routes with the extended score. The gap in the amount of traffic passing by the routes with slope $\in$ [1–9%] is less than 0.5%; the gap is around 1% for the steepest (slope > 9%) and downhill (slope < 3%) routes. These differences in traffic are mainly the consequence of avoiding the steepest routes with the extension module.

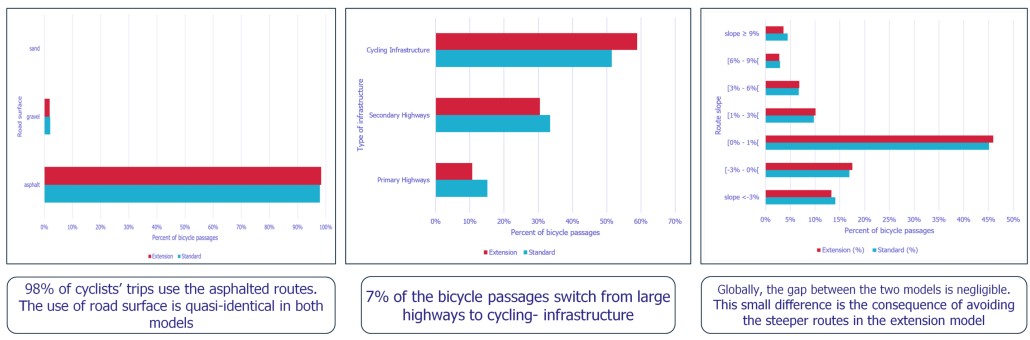

**Figure 7.** MATSim bicycle standard (blue) and extension (red) models: extraction of simulation results.

Figure 8 shows the simulated cycling flows with both approaches. With the standard algorithm, an important flow is passing by the departmental routes. They are located in densely populated areas and host numerous facilities. With the extended algorithm, the flow is reduced in the departmental routes due to the negative marginal utility of using large highways and is reinforced in the cycleways.

Table 4 presents the simulated activity for the standard model; for a better explanation, the trips are classified into short, intermediate and long trips. Short trips are the most frequent (85% of trips). The average travel distance is around 1 km for 6 min travel time and 16 m positive elevation. Intermediate trips represent 10% of trips, for 5 km distance, 17 min travel time and +70 m elevation. Long trips represent 5% of bicycle trips. The average distance is around 7 km, 25 min travel and 100 m positive elevation. Globally, the average travel distance is around 2 km, 7 min travel time and 26 m positive elevation.

**Table 4.** Bicycle trips: positive elevation for short, intermediate and long trips.

| Range (km) | [0–4] | [4–6] | [6–10] | All |
|---|---|---|---|---|
| Average (m) positive elevation | 16.4 | 70.3 | 99.8 | 26.1 |

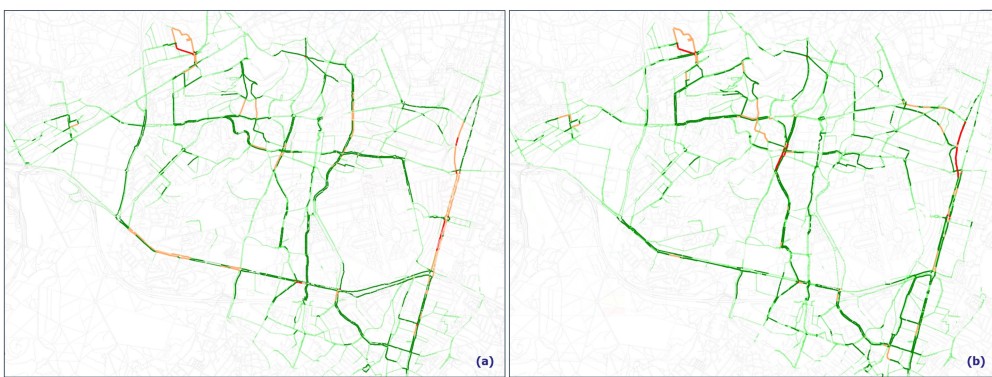

**Figure 8.** Main bicycle flows in standard scenario (**a**), and the extension scenario (**b**).

Figure 9a illustrates the variation of the average trip distance by trip duration. It shows that the travel time and distance are almost equal for the short trips. For the intermediate trips, the differences are not significant. The travel distance is increased by 0.7% (30 m), while the travel time is increased by 2.5% (30 s). For the longest trips, the differences are more important. The extension scenario trips are longer by 6% (400 m) in travel distance and 8% (2 min) in travel time.

Figure 9b depicts the evaluation of the average trip distance by the average positive elevation. The difference in positive elevation is negligible for trips shorter than 2 km (less than 2 m). It is briefly decreased for trips shorter than 4 km by 14% (8 m), and the gap is decreased linearly for the intermediate trips by 25% (17 m) and by 24% for the longest trips.

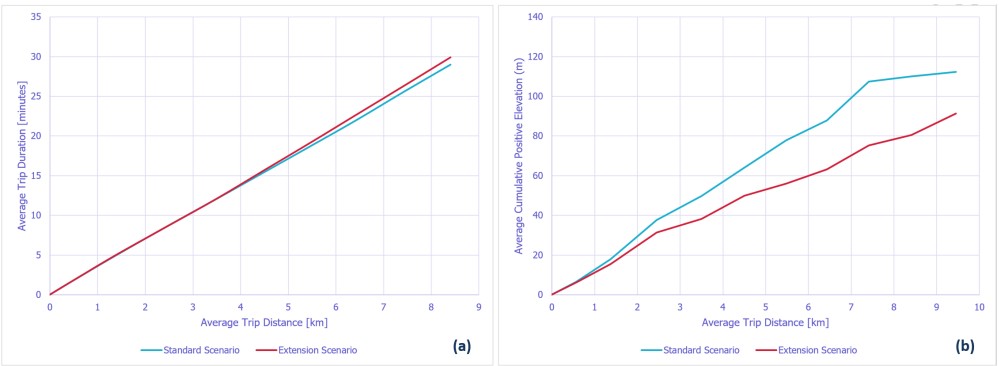

**Figure 9.** Average travel duration (**a**) and travel positive elevation (**b**) by travel distance.

### 5.3. Elevation Imprecision: Sensitivity to Gradient Estimation

Nodes and links in OpenStreeMap data do not include the altitude or slope information. In absence of a terrain survey done specifically for an area of study, one has to use a digital elevation model. As pointed out in [26], there are two types of digital elevation model: digital surface model created from satellite imagery, and digital terrain model from aerial photogrammetry. In an urban environment, a digital surface model is not filtered with the altitude of the bare road: its includes buildings and trees elevation. A digital terrain model represents the ground surface of the Earth with no objects, but this kind of data is usually not publicly available. The European Digital Elevation Model EU-DEM used by default in the extended bicycle model of MATSim and in the study of Ziemke et al. [4] is a post-processing of a combination of high resolution digital surface models from the NASA, where a broad range of artifacts have been removed.

The EU-DEM database resolution is 25 m, meaning that the continental area is divided $25 \times 25$ m cells: all points within a cell have the same estimated altitude from the sea level. By using this public database to estimate the elevation of a road network, the nodes located in the same mesh have the same Z-coordinate. It induces an error in the slopes estimation: a link connects two nodes located in the same cell has an estimated slope of 0%, which can be unrealistic. On the other hand, the slope of a link crossing the frontier of a cell may be overestimated as the altitude of its two terminal nodes are from two different cells. It can generate unrealistic behavior of the cyclist: for instance, the velocity can be suddenly decreased from 20 km/h to 3–4 km/h for a few meters then back to 20 km/h in.

This problem occurs especially when the two nodes are far from the center of their cell: it can be addressed for short length road segments. Statistics on our network are as follow: 77% of network links are shorter than $25\sqrt{2} = 35$ meters, and 0% slope links represent 36% of the total number of bicycle links. Figure 10 illustrates the distribution of the links slopes estimated with the original EU-DEM data in blue bars. As can be seen, the 0% slopes are predominant because of the rather large part of short-length links in this area.

To estimate the global error due to the elevation precision, we perform a sensitivity analysis. Two bike travel simulation are run and compared: the original approach of Ziemke et al. with no further processing of the EU-DEM data, and the same algorithm with smoothed link slopes.

The smoothing of the links slope that we perform adjusts the elevation coordinates of the network node according to the following:

- Elevation of the surrounding cells;
- Relative distance between the selected node and its grid cell center, and the surrounding cell centers.

To go into detail, let us call $C$ the set of surrounding cells of a given node $nd$ belonging to a cell $C_0$. Set $C$ excludes cell $C_0$. Let us call $C_1$ the cell of set $C$ that is the closest to node $nd$ estimated by computing the distances from node $nd$ to the center of each cell in set $C$. Then the altitude coordinate of node $nd$ is expressed as

$$z_{nd} = z_{C_0} + \alpha(z_{C_1} - z_{C_0}) \text{ with } \alpha = \frac{\|nd - C_0\|}{\|C_1 - C_0\|} \tag{4}$$

The result of this linear filtering is illustrated in Figure 10, showing the distribution of slopes after processing with red bars. As can be seen, the original peak of the small slopes is re-balanced mainly in the range $[-3\%, +3\%]$, providing a more realistic distribution of slopes over this area.

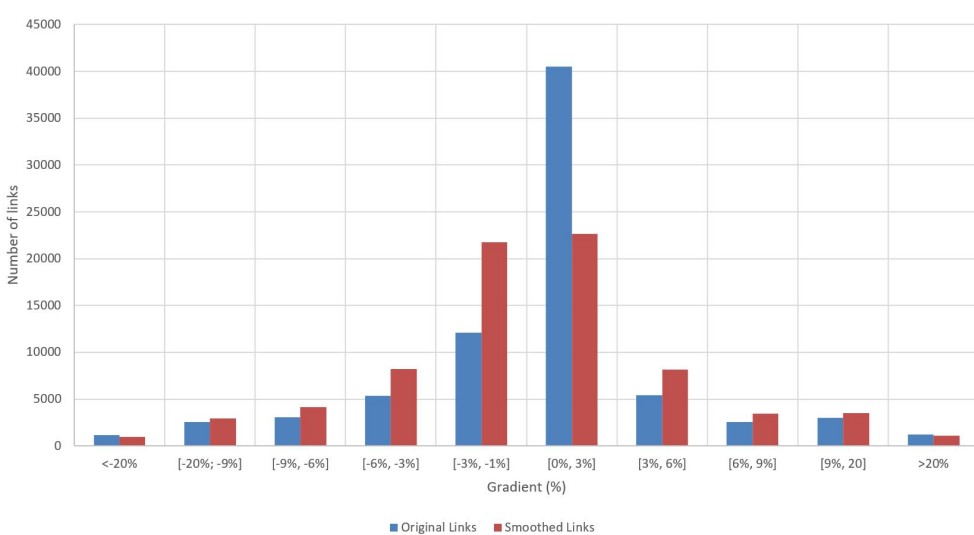

**Figure 10.** Network links gradient distribution: original and smoothed estimation of slopes.

We run the two simulations of bicycle travels with the same synthetic population generated described in Section 5.1. One simulation with the original links, the other with smoothed slopes. To compare the output of these simulations, we use two metrics: travel distance and travel time. In Figure 11, the difference between these metrics is plotted as a distribution for a range of travel distance. The difference is expressed as a deviation between these values expressed in percent. For instance, for trips shorter than a kilometer, the difference in computed travel distances with the two networks is 0.4%, whereas the travel times differ only by 0.8%. Overall, one can see from this figure that the low resolution of the altitude data impacts minimally the output of the extended algorithm proposed by Ziemke et al.

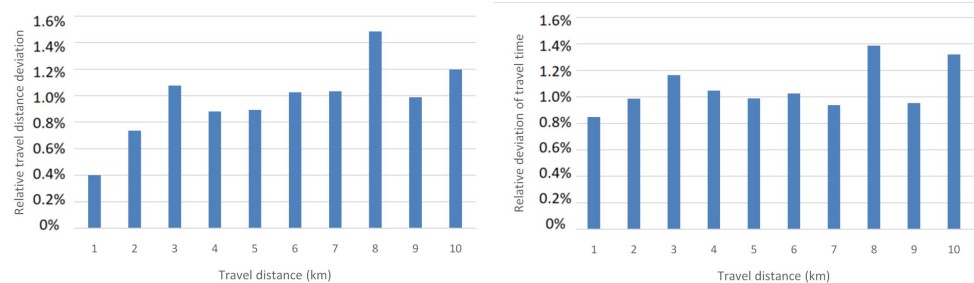

**Figure 11.** Comparison of simulations of bicycle travels with and without slope correction. Travel distance (**left**) and travel time (**right**) difference is minimal, even for long trips.

*5.4. Hypothesis and Limitations*

Some limitations and hypotheses were made and can be listed as follows. One of the limitations of the algorithm is that it cannot extend its modeling to patterns of mobility that are not already and specifically sensed.

- The proposed model does not consider socio-economic variables (age, employee status, etc.) in the modal choice process. For example, this study does not incorporate information on agents' age, gender, weight, or health.
- The model cannot infer activity patterns that are absent in the data, such as the mobility of people who go back home for lunch or those who go shopping during the lunch break, for instance.
- The typical day modeled is a day of the week in a pre-COVID-19 world, and the adaptation to a post-COVID-19 scenario requires time distributions that are still unknown at the time of publication. In a post-COVID-19 situation, the time of activities, especially work hours, can be expected to change with the introduction of a work-from-home and more agile office model and, therefore, the time distribution of connected secondary activities. More importantly, the share of bicycle commuters in transportation can be expected to grow significantly.

*5.5. Discussion*

The results show that road characteristics' impact depends on the trip type. For short trips, the selected routes are often residential and living-street highways, with no need to pass by large highways. A short trip means a short travel time and less variation in the geographical area, which reduces the impact of gradient and type of route. The large routes and cycleways propose a fast and continuous infrastructure for the cyclist in the standard scenario for long trips. These highways are penalized by an important reduction of travel speed in the extended algorithm, which impacts the cyclist's route choice and forces them to switch to cycleways and residential routes: this switch increases the use of cycling infrastructure compared to the standard algorithm.

The road gradient impacts long trips more than short trips: a long trip has an important cumulative positive elevation, which significantly reduces its score compared to a short trip. Avoiding the steep routes leads to a considerable decrease in cumulative positive elevation in the extension scenario compared to the standard scenario.

In the current study area, the impact of the gradient is not very significant for the following reasons. The large part of trips are short, the proportion of steep routes is very low, and it is concentrated mainly in one continuous zone. Only the passage through this zone will impact the cyclist's route choice. The impact of the gradient on travel speed depends mainly on the route slope. In the extended algorithm, the speed reduction is linear with the route slope, which means that only passages by very steep sections will reduce the velocity significantly.

Another aspect to be considered when dealing with road gradient is the resolution of the elevation data of EU-DEM [23]: elevations are sampled every 25 m, but many links are smaller than that. If used as is, the extension model can lead to a large sampling error in

calculating road slopes with some spikes in the gradient estimation. We showed, however, that the overall impact of the small resolution of the elevation data is minimal in terms of travel distance and travel time, at least in the study area.

As recommendations for practitioners of transport modeling willing to refine their bicycle commuting analyses, several aspects can be highlighted:

- The need for a refined slope estimation module: the gradient output by the Ziemke module can be used as is if the region of interest is flat. Otherwise, some algorithm modification to compute the gradient is required, or a finer resolution of the altitude map.
- The MATSim extension can be used for motored personal mobility devices: in the case of electric bikes or e-scooters, the dis-utility factors must be adapted. Indeed, e-bikes are less subject to speed reduction and can easily use primary and secondary highways because their speed is 25 km/h on flat ground. On ascending slopes, their speed depends on their power and autonomy: for some 24 V/36 V low-cost e-bikes, the speed can be reduced to 10 km/h on a 10% slope, whereas a high-end 48 V scooter could easily maintain 25 km/h or more.

## 6. Conclusions

In this paper, we evaluated the benefits of using a refined bicycle model that considers infrastructure characteristics compared to a standard model that assumes flat terrain and homogeneous road. We experimented with the two approaches on a hilly relief in a large suburban area with a particularly challenging altitude profile and a large variety of roads, whether in type—from residential to national highway—or in the pavement surface. The type of infrastructure included primary, secondary and residential highways and residential and cycling ways. Pavement surface was made of asphalt, gravel and sand, a specificity of the suburban area with many green spaces, such as parks and forests.

We implemented a method for synthetic population generation with a focus on activity–time choice, location choice and modal choice at a regional scale to construct plannings for 126k agents over five municipalities, with chains of activity made of work, education, shopping, leisure, restaurant and kindergarten, which fit adequate, real-world time distributions. We focused on the cyclists transportation model: one standard model where route choice and travel time are calculated with a 2D flat Earth hypothesis, and an extended model that considers road type, pavement surface and gradients. Our finding is that MATSim bicycle extension model impacts mainly the long trips. The differences between the two models are minimal in terms of travel time and distance. The impact of pavement surface is negligible when comparing the two approaches. The impacts of road slope are very limited: only the long trips are impacted. Mainly the type of infrastructure impacts the cyclist's route choice: the refined model makes cyclists switch from large highways to cycling infrastructure.

For the practitioners, two conclusions can be drawn from our study. First, a refined bicycle transportation model does not change mobility patterns significantly compared to a flat Earth simulation. It can be re-insuring for a multimodal traffic model where a high precision of bicycle traffic flow is not required. Second, if a study focused on bicycles aims at estimating or forecasting their flows very precisely, a careful gradient estimation must be processed.

An interesting perspective of this work would be considering the simulation of electric bikes. Their specificity would require further modifications of the algorithm to take into account the power and autonomy of the vehicles and the propensity of such users to use the cars road network. This is especially needed when the demand for longer trips by bike is increasing and encouraged for sustainability purposes.

**Author Contributions:** Investigation, Y.D. and R.B.; Methodology, Y.D., R.B. and F.D.; Supervision, R.B.; Writing—original draft, Y.D. and R.B.; Writing—review and editing, R.B., F.D. and M.Z. All authors have read and agreed to the published version of the manuscript.

**Funding:** This research was funded by the E3S project, a partnership between Eiffage and the I-SITE FUTURE consortium. FUTURE bénéficie d'une aide de l'État gérée par l'Agence Nationale de la Recherche (ANR) au titre du programme d'Investissements d'Avenir (référence ANR-16-IDEX-0003) en complément des apports des établissements et partenaires impliqués.

**Institutional Review Board Statement:** Not applicable.

**Informed Consent Statement:** Not applicable.

**Data Availability Statement:** Not applicable.

**Conflicts of Interest:** The authors declare no conflict of interest.

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
