# Peer review of "Analysis of MATSim Modeling of Road Infrastructure in Cyclists’ Choices in the Case of a Hilly Relief"

_infrastructures, doi:10.3390/infrastructures7090108_

Round 1
Reviewer 1 Report
The authors presented a manuscript that describes research on cyclists’ route choice based on MATsim taking the street grade into account.
The manuscript has two important issues. First, several aspects of the experiment are not clearly explained. Second, the initial assumptions for the experiment might have led to flawed outcome, which may call into question the results obtained.
Line 8: the abstract should contain a brief description of the research results and conclusion.
The introduction is meagre and many studies that addressed similar problems are missing. Below is a non-exhaustive list of documents that may enhance the literature review.
Casello, J. M., & Usyukov, V. (2014). Modeling cyclists’ route choice based on GPS data. Transportation Research Record, 2430(1), 155-161.
Chen, C. F., & Chen, P. C. (2013). Estimating recreational cyclists’ preferences for bicycle routes–Evidence from Taiwan. Transport Policy, 26, 23-30.
Yang, C., & Mesbah, M. (2013, October). Route choice behaviour of cyclists by stated preference and revealed preference. In Australasian Transport Research Forum 2013 Proceedings.
Lines 28 and 33: sentences should make sense when removing the square brackets. Include author’s name or rephrase the sentence.
Line 45: The authors state that a synthetic population was generated. If no real data were used, then what is the usefulness of the study?
Line 50: If no mixed traffic conditions are considered, the findings of the study are very limited.
Line 75: What do you mean by “resolution”? Height precision, horizontal grid spacing, other? This might be a severe limitation of the study.
Line 124: data presented in table 3 are not clearly explained. What are these values and where do they come from? What are F_inf and F_conf?
Line 139: How was the “catchment area” selected? What is its size and the total length of links?
Line 140: Where do these trip data come from?
Line 146: How where these grades determined?
Line 233: the “conclusions” section is also meagre, indicating that the usefulness of the study is little or negligible.
Reviewer 2 Report
The article concerns an interesting topic which is "Analysis of MATSIM Modeling of Road Infrastructure in Cyclists' Choices in the Case of a Hilly Relief". The article concerns the current subject, both in terms of research and application. I only have a few comments about it.
First, the abstract is quite short. I would propose to extend it with broader deliberations on the theoretical outline and a sentence about the results obtained. It would also be nice to indicate more keywords.
In the introduction, I don't like your starting a sentence with eg [4]. If you want to refer to a given reference, point out that the given author X claims something ...
I also think the introduction is too short. Reference should be made to the current situation regarding bicycles, to the current research (Matyja, T .; et al. (2022) Possibility to Use Professional Bicycle Computers for the Scientific Evaluation of Electric Bikes: Velocity, Cadence and Power Data) and to market trends such as the growing bike-sharing systems (Czech, P. et al. (2018). Bike-Sharing as an Element of Integrated Urban Transport System). I am showing you valuable literature to which please refer in the text.
The methodological and experimental part is correct in my opinion.
The discussion does not refer to research carried out by other scientists - please complete it by pointing to the differences or similarities related to the topic under study.
The summary lacks specific conclusions in the form of recommendations that could be applied, for example, by practitioners or policy makers. It would also be interesting to indicate whether your conclusions can be referred to, for example, other forms of micro-mobility and how they could improve it, or how you can reduce various types of problems, such as, for example, with scooter-sharing. (please refer to Turon et al. The concept of rules and recommendations for riding shared and private e-scooters in the road network in the light of global problems).
In summary, please indicate the limitation of your method and future research.
After completing the indicated elements, the article will be ready for publication. Good luck!
Reviewer 3 Report
The paper presents an interesting case study developing the methodology of bicycle journey modeling. Consideration of road characteristics and the use of effective tools (Matsim) give useful results.
The study area should be more detailed and described. What five suburban cities are considered? In which location?
The weak side of the manuscript is a very poor literature review. I identified in "References" only 3 scientific papers dedicating “bicycle traffic” (I did not consider the Authors self publications). These publications are quite old (2010, 2014, 2019). The literature concerning “bicycling” is very rich. For example, the “Science Direct” search gives 19,151 results for the keywords “bicycle traffic”. The reviewed manuscript is not a pure literature review, but I strongly recommend the addition of a few characteristic and representative sources from actual scientific papers. This will make a better background for the presented research. So, the separate section “Literature review” is necessary. Broader comments to this review will be nice in the “Discussion part” of the paper.
Apart from the above, I formulated some small, more technical remarks.
The sentences shouldn't start with “[4]” or “[5]” (page 1). The better form will be “Article [4]…” or “The study [5]…”.
Tables 1 and 2 were not commented on in the text.
The title of figure 3 is too long. The title should not contain the comments. Please shorten this title and transfer the commentary to the text.
Round 2
Reviewer 1 Report
The authors have somewhat improved the manuscript in the revised version. However, there are a few issues that must be addressed before it is suitable for publication.
There are some sentences in the text added in the last version that are unclear, a fact that reduces the readability. The manuscript would benefit from proofread.
The stated elevation precision (25 m) may be really poor for estimating street longitudinal grades. The authors should perform a sensitivity analysis to quantify the estimation error.
Figure 2 should be split into two figures. The elevation profile in the second one is hardly readable and its direction is inverted with respect to the orientation of the map.
Reviewer 2 Report
The article concerns an interesting topic which is "Analysis of MATSIM Modeling of Road Infrastructure in Cyclists' Choices in the Case of a Hilly Relief". The article concerns the current subject, both in terms of research and application. I only have a few comments about it.
First, the abstract is quite short. I would propose to extend it with broader deliberations on the theoretical outline and a sentence about the results obtained. It would also be nice to indicate more keywords.
In the introduction, I don't like your starting a sentence with eg [4]. If you want to refer to a given reference, point out that the given author X claims something ...
I also think the introduction is too short. Reference should be made to the current situation regarding bicycles, to the current research (Matyja, T .; et al. (2022) Possibility to Use Professional Bicycle Computers for the Scientific Evaluation of Electric Bikes: Velocity, Cadence and Power Data) and to market trends such as the growing bike-sharing systems (Czech, P. et al. (2018). Bike-Sharing as an Element of Integrated Urban Transport System). I am showing you valuable literature to which please refer in the text.
The methodological and experimental part is correct in my opinion.
The discussion does not refer to research carried out by other scientists - please complete it by pointing to the differences or similarities related to the topic under study.
The summary lacks specific conclusions in the form of recommendations that could be applied, for example, by practitioners or policy makers. It would also be interesting to indicate whether your conclusions can be referred to, for example, other forms of micro-mobility and how they could improve it, or how you can reduce various types of problems, such as, for example, with scooter-sharing. (please refer to Turon et al. The concept of rules and recommendations for riding shared and private e-scooters in the road network in the light of global problems).
In summary, please indicate the limitation of your method and future research.
After completing the indicated elements, the article will be ready for publication. Good luck!
Round 3
Reviewer 1 Report
The authors have made a nice effort to enhance the manuscript, acknowledging the main limitations and improving English language to some extent. With the available data, as stated by the authors, and the initial approach, there is little they can do to improve further the manuscript. Given that Infrastructures is a non-JCR journal, the manuscript could be accepted.
Reviewer 2 Report
The article has been significantly improved and is suitable for publication in its current form. Thank you to the authors for the changes made.
Author Response
The authors would like to thank you very much for your valuable remarks and corrections, which we hope participate in improving the paper quality and possible impact.